

# Influence of engaging female caregivers in households with adolescent girls on adopting equitable family eating practices: a quasi-experimental study

Hanna Gulema[1], Meaza Demissie[1], Alemayehu Worku[2],
Tesfaye Assebe Yadeta[3] and Yemane Berhane[4]

[1] Department of Global Health and Health Policy, Addis Continental Institute of Public Health, Addis Ababa, Ethiopia
[2] School of Public Health, Addis Ababa University, Addis Ababa, Ethiopia
[3] School of Nursing and Midwifery, Haramaya University, Harar, Ethiopia
[4] Department of Epidemiology and Biostatistics, Addis Continental Institute of Public Health, Addis Ababa, Ethiopia

Corresponding author
Hanna Gulema,
hannagulema@addiscontinental.edu.et

## ABSTRACT

**Background:** In patriarchal societies, female caregivers decide on food allocation within a family based on prevailing gender and age norms, which may lead to inequality that does not favor young adolescent girls. This study evaluated the effect of a community-based social norm intervention involving female caregivers in West Hararghe, Ethiopia. The intervention was engaging female caregivers along with other adult influential community members to deliberate and act on food allocation social norms in a process referred to as Social Analysis and Action (SAA).
**Method:** We used data from a large quasi-experimental study to compare family eating practices between those who participated in the Social Analyses and Action intervention and those who did not. The respondents were female caregivers in households with young adolescent girls (ages 13 and 14 years). The study's outcome was the practice of family eating together from the same dish. The difference in difference (DID) analysis with the mixed effect logistic regression model was used to examine the effect of the intervention.
**Result:** The results showed improved family eating practices in both groups, but the improvement was greater in the intervention group. The DID analysis showed an 11.99 percentage points greater improvement in the intervention arm than in the control arm. The mixed-effect regression produced an adjusted odds ratio of 2.08 (95% CI [1.06–4.09]) after controlling selected covariates, *p*-value 0.033.
**Conclusions:** The involvement of influential adult community members significantly improves the family practice of eating together in households where adolescent girls are present in our study. The intervention has great potential to minimize household food allocation inequalities and thus improve the nutritional status of young adolescents. Further studies are necessary to evaluate the effectiveness of the intervention in different social norm contexts to formulate policy and guidelines for scale-up.

# INTRODUCTION

Improving adolescent girls' nutritional status is a critical milestone in achieving the Sustainable Development Goals (*United Nations, 2015*). Eating together as a family has many positive outcomes, including equitable food distribution and improved nutritional and health status in low-income settings. Families who eat together tend to have a more equitable food distribution, with all members receiving a fair share of available food (*Lee et al., 2014*).

Though eating together as a family has beneficial health outcomes, family eating practices are often influenced by prevailing social norms, making it challenging to promote the equable distribution of food within the household (*Monterrosa et al., 2020*). Social norms often dictate food consumption, including who, when, and how much to eat. Influential adult community members and caregivers often enforce the norms (*Blum et al., 2019*; *Baird et al., 2019*). Within households, female caregivers have a more significant role in food preparation and allocation, often guided by the prevailing social norms (*Corley et al., 2022*). In low-income countries, discriminatory social norms often make young adolescent girls eat less frequently or less, especially during periods of scarcity (*Monge-Rojas et al., 2021*; *Nurmawati, Rachmawati & Muna, 2022*).

Food allocation social norms that lead to unequal consumption can severely affect adolescent girls' food security and nutritional status (*Blum et al., 2019*). In addition, such social norms can lead to insufficient consumption to meet daily dietary requirements resulting in poor mental and physical health outcomes (*Alonso, 2014*; *Das & Mishra, 2021*). Adolescent girls with depleted nutrients may experience fatigue, weakness, and difficulty concentrating in school, limiting their potential and opportunities in their educational and career development (*Belachew et al., 2011*; *Monge-Rojas et al., 2021*).

When families eat together, everyone around the table has a better opportunity to share the available food fairly, even if there is a shortage in the household (*Monge-Rojas et al., 2021*). This is especially important for vulnerable groups like adolescent girls, who might otherwise be left with little or nothing unless they eat together with other family members (*Sharps & Robinson, 2017*; *Chattopadhyay et al., 2022*). To improve equitable household food distribution practices for adolescent girls, breaking the food inequality cycle resulting from gender-biased social norms is crucial (*Liu et al., 2022*).

Social Analysis and Action (SAA) promotes community dialogue among the people who influence the norms (*e.g.*, community leaders, religious leaders, mothers-in-law, and husbands) to shape existing expectations, decisions, and behaviors around household food allocation. For the purpose of this study, the community dialogue centered around adolescent girls' nutrition. SAA is regarded as a community-led social change process. In this approach, community members are active and central in identifying social issues, setting goals, and implementing positive social changes (*Yimenu, 2020*; *Chowdhary et al., 2022*). Previous research has suggested that simple information or education about healthy

eating practices may not be sufficient to change social norms and promote behavioral changes. Instead, targeted interventions that address specific social norms are more effective in promoting behavior change (*Khalid, Gill & Fox, 2019*). The engagement of influential norm holders and gatekeepers is critical to bring about sustainable and acceptable changes in social norms. Nevertheless, research on social norms related to food allocation inequality is scarce in Ethiopia. Therefore, this study assesses the effect of engaging norm influencers on improving family eating practices.

## MATERIALS AND METHODS

### Study setting, design, and population

The study was conducted in the West Hararghe zone of the Oromia regional state in Ethiopia. The study was conducted in three Woredas, where two weredas implemented SAA intervention and one Wereda served as a comparison area. The majority of the population in these areas belongs to the Oromo ethnic group and follows the Islam faith. In addition, agriculture-based living styles are dominant in the area.

This article utilized data from a large quasi-experimental study that assessed various interventions to address structural issues affecting young adolescent girls in the Western Hararghe Zone, Oromia Region, Ethiopia (*Berhane et al., 2019*). This study's population was female caregivers with at least one young adolescent girl (age 13 and 14) in the household. The intervention primarily targeted female caregivers who are responsible for food preparation and allocation in the households.

### Intervention (SAA)

The SAA was initiated by assembling female caregivers along with other influential adults in the community, including religious leaders and other norm holders. The group consists of about 30 influential adults to ensure adequate dialogue and greater impact by engaging influential adults in the process, which promotes open and candid discussion, seeking solutions to identified challenges, and promoting actions to address the challenges. The process starts with the training of the partcipants by trained facilitators. The SAA was facilitated by selected and trained community members from each group. Trained support staff were assigned to support the groups in person or *via* telephone to maintain fidelity to the intervention. The SAA is designed to change community social norms through open dialogue (*Tasew et al., 2018*). The groups met monthly, and each discussion session lasted an hour or more. The central discussion topics girls related to this research outcome were adolescent girls' nutrition. The group met every month for 3 years during the intervention period.

### Mechanisms/causal pathways

The intervention (SAA) influences the behavior of individuals or groups, leading to changes in their decision-making, habits, or actions related to family eating practices, focusing on adolescent girls. For instance, understanding adolescent girls' nutritional needs and correcting cultural miss conception might lead to changes in intrahousehold

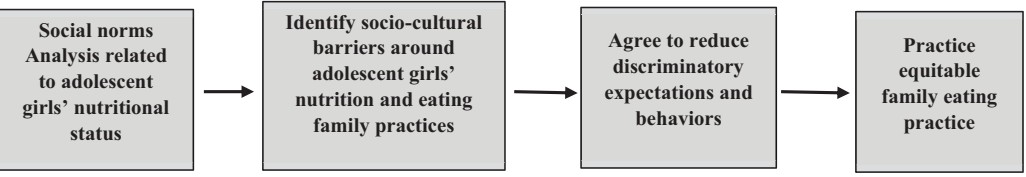

**Figure 1 Theory of change model for community-level intervention (Social Analysis and Action).** Theory of change model shows how community-level intervention (Social Analysis and Action) leads to change in household level eating practice.

family eating practices. In addition to this, Socio-cultural contexts can play a role in how individuals or groups behave in a given situation. These factors might mediate or modify treatment effects (Fig. 1).

## Sampling procedure

The study population was selected by a two-stage cluster sampling method, as it was described in the previous article (*Berhane et al., 2019*). First, for each study arm (intervention and comparison arms), 38 clusters (locally known as development areas or 'Gere,' a division within the smallest administrative unit, the kebele) were randomly selected after preparing the list of all clusters in the study woredas. Then, in each cluster, the list of households with adolescent girls (aged 13–17 years) was prepared by conducting a complete household listing. In the final stage, 30 households with eligible adolescent girls were randomly selected from the list. Then, female caregivers in the sampled households were invited to participate in the study.

## Data collection

A structured questionnaire was used to collect relevant data. The questionnaire contains questions related to socio-demographic variables from the Ethiopian Demographic and health survey (EDHS) (*Ethiopian Public Health Institute (EPHI) [Ethiopia] and ICF, 2021*) and standard questions used for the household food security assessment (*Coates, Swindale & Bilinsky, 2007*). The questionnaire was first developed in English and then translated into the local language, Afaan Oromo. Data were captured before the intervention (at baseline) and after a period of implementing the intervention for 3 years (at end line) using an open data kit (ODK), which is an electronic data collection platform. Data were collected by trained enumerators fluent in the written and spoken local language, Afaan Oromo.

## Measurement

The effect of the intervention (social analysis and action) was measured by the number of families eating together, which was assessed at baseline before the intervention commenced and at the end-line after the intervention. The family was considered as eating together if a family mostly ate at the same time from the same dish, as reported by female caregivers. Then, the family eating practice, the outcome variable, was dichotomized into eating together or not.

## Data analysis

The data analysis was done on a combined baseline and end-line dataset. A time variable was created as 'time,' the baseline dataset was coded as '0,' and the endline was coded as '1'. We compare differences in eating practices between the study groups using chi-square tests. The Difference-in-Difference (DID) approach is a method to assess the program's impact by comparing changes over time between the comparison group. For this study, the DID model is based on a mixed-effect logistic regression. The model assumes that in the absence of the intervention, the difference between the intervention and comparison group would be constant over time, which is a parallel trend assumption. Difference-in-Difference is implemented by adding an interaction term between time and program group dummy variables in a regression model and can be specified as follows:

$$Y_i = b_0 + b_1 * [Time_i] + b_2 * [program_i] + b3 * [Time_i * Program_i] + b_4 * [Covariates] + \varepsilon_i$$

in which $y_i$ is the outcome variable, which is family eating practice measured at the baseline and end line in both intervention and comparison group.

$\beta_0$ is the intercept, which is the value of the outcome variable when all of the other variables are equal to zero.

$Time_i$ is a time variable, which is the baseline or end-line period that takes the value 0 or 1 respectively.

$Program_i$ is a program variable, which is the control or treatment groups that coded as 0 or 1 respectively.

$(Time_i * program_i)$ is an interaction term between time and program, which used to test the effect of the intevention.

$\varepsilon_i$ is the error term, which captures the effect of all factors that affect the outcome but the model could not adequately represent.

The models were adjusted for clustering and unbalanced covariates between the intervention and control group. A mixed-effect model was used to control the clustering effect as the data is clustered. As the outcome variable was binary, we used mixed-effect logistic regression. We used the Difference-in-Difference (DID) model with the interaction terms to determine the intervention effect. The impact was examined based on a statistically significant difference of $p$-value < 0.05, the crude odds ratios (COR), and adjusted odds ratios (AOR) with 95% confidence intervals (CI) of the interaction term (program × time), using Stata version 14 statistical software. We checked for multicollinearity and outliers during the analysis. Mixed effect logistic regression controlled for clustering effect and other variables such as female caregiver's educational status and income, male caregivers' education status and income, household wealth and food security status, intervention, time, and difference in the different interaction terms.

## Ethical considerations

The study protocol was reviewed and approved by the Ethical Review Board of the Addis Continental Institute of Public Health (Ref No. ACIPH/IRB/005/2016). Due to the low literacy level of the community and the minimal risk involved in the study, informed verbal consent was obtained after explaining the purpose of the study. Participants were

**Table 1 Background characteristics of female caregivers, West-Hararghe, Ethiopia.**

| Background characteristics | Baseline | | End line | |
|---|---|---|---|---|
| | Intervention arm<br>n (%) | Control arm<br>n (%) | Intervention arm<br>n (%) | Control arm<br>n (%) |
| **Male caregiver's educational status** | | | | |
| Never attended school | 199 (51.96) | 182 (47.52) | 223 (51.98) | 231 (43.58) |
| Primary & above | 184 (48.04) | 201 (52.48) | 206 (48.02) | 299 (56.42) |
| **Female caregiver educational status** | | | | |
| Never attended school | 306 (79.90) | 321 (83.81) | 356 (82.98) | 419 (79.06) |
| Primary & above | 77 (20.10) | 62 (16.19) | 73 (17.02) | 111 (20.94) |
| **Adoelscent girls education** | | | | |
| Never attended school | 52 (13.58) | 42 (10.97) | 12 (2.80) | 11 (2.08) |
| Primary & above | 331 (86.42) | 341 (89.03) | 417 (97.20) | 519 (97.92) |
| **Male caregivers having income** | | | | |
| Yes | 278 (72.58) | 225 (58.75) | 280 (65.27) | 408 (76.98) |
| No | 105 (27.42) | 158 (41.25) | 149 (34.73) | 122 (23.02) |
| **Household wealth tercile** | | | | |
| First | 113 (29.50) | 102 (26.63) | 181 (42.19) | 163 (30.75) |
| Second | 138 (36.03) | 161 (42.04) | 120 (27.97) | 160 (30.19) |
| Third | 132 (34.46) | 120 (31.33) | 128 (29.84) | 207 (39.06) |
| **Household food security status** | | | | |
| Insecure | 308 (80.42) | 308 (80.42) | 257 (59.91) | 390 (73.58) |
| Secure | 75 (19.58) | 71 (18.54) | 172 (40.09) | 140 (26.42) |

**Note:**
The data shows background characteristics of female caregivers.

informed about their rights to refuse participation and/or withdraw their consent at any time. All interviews took place in private settings. No personal identifiers were linked with the dataset made available for this study.

## RESULT

A total of 812 caregivers in the intervention arm and 913 in the control arm were included in the study. The mean ± Standard Deviation age of female caregivers was 39.85 ± 12.39 in the intervention arm and 39.40 ± 10.87 in the control arm. Most respondents in both arms had primary school education and were ever married, Muslim, and from food-insecure households (Table 1). The baseline balance for covariates showed no significant differences between the comparison groups except for male caregiver income (Table 2).

### The effect of the intervention

The family eating practice, eating together, improved in both study arms but more in the intervention arm. There was a 20.57% increase in the intervention arm compared to 8.58% in the comparison group. Overall, the intervention group showed an 11.99% increase compared to the comparison group (Pr = 0.001) (Table 3).

**Table 2 Baseline characteristics balance between treatment and control, West-Hararghe, Ethiopia.**

| Background characteristics | Baseline | | Chi-square test (p < 0.05) |
| --- | --- | --- | --- |
| | Intervention arm n (%) | Control arm n (%) | |
| **Male caregiver's educational status** | | | 0.22 |
| Never attended school | 199 (51.96) | 182 (47.52) | |
| Primary & above | 184 (48.04) | 201 (52.48) | |
| **Female caregiver educational status** | | | 0.16 |
| Never attended school | 306 (79.90) | 321 (83.81) | |
| Primary & above | 77 (20.10) | 62 (16.19) | |
| **Adolescent girl's educational status** | | | 0.271 |
| Never attended school | 52 (13.58) | 42 (10.97) | |
| Primary & above | 331 (86.42) | 341 (89.03) | |
| **Male caregivers having income** | | | 0.007 |
| Yes | 278 (72.58) | 225 (58.75) | |
| No | 105 (27.42) | 158 (41.25) | |
| **Household wealth tercile** | | | 0.23 |
| First | 113 (29.50) | 102 (26.63) | |
| Second | 138 (36.03) | 161 (42.04) | |
| Third | 132 (34.46) | 120 (31.33) | |
| **Household food security status** | | | 0.71 |
| Insecure | 308 (80.42) | 308 (80.42) | |
| Secure | 75 (19.58) | 71 (18.54) | |

**Table 3 Change between baseline and end line of eating together practice between the comparison arms, West-Hararghe, Ethiopia.**

| Scale | Intervention arm | | | Control arm | | | DID | |
| --- | --- | --- | --- | --- | --- | --- | --- | --- |
| | Baseline # (%) | Endline # (%) | Difference (%) | Baseline # (%) | Endline # (%) | Difference (%) | DID (%) | Chi-square test |
| **Family eating practice** | | | | | | | | |
| Family eat together | 73 (19.06) | 206 (39.63) | 20.57 | 77 (20.10) | 158 (28.68) | 8.58 | 11.99 | p = 0.001 |
| The family did not eat together | 310 (80.94) | 259 (60.37) | | 306 (79.90) | 378 (71.32) | | | |

**Note:**
Each data point shows change between baseline and end line of eating together practice between the comparison arms, West-Hararghe, Ethiopia.

After controlling for potential confounders and clustering effect, the DID analysis showed a significant difference in eating together between the comparison groups. The adjusted odds ratio showed that eating together was twice as likely in the intervention arm as compared to the control arm (AOR 2.08 (95% CI [1.06–4.09]), p-value of 0.033). In addition, male caregivers' education, income, and household food security status were significantly associated with the outcome (Table 4).

**Table 4 Estimating the effect of the interventions on family eating practice after adjusting for other variables, West-Hararghe.**

| | Crude odds ratio (95% confidence interval) | Adjusted odds ratio (95% CI) | p-value |
|---|---|---|---|
| **Program** | | | |
| Control arm | 1 | 1 | |
| Intervention arm | 1.34 [0.93–1.93] | 0.91 [0.55–1.52] | 0.73 |
| **Time** | | | |
| Baseline | 1 | 1 | |
| Endline | 2.18 [1.54–3.09] | 1.39 [0.87–2.24] | 0.17 |
| DID Interaction [Time * program] | 2.43 [1.65–3.57] | 2.08 [1.06–4.09] | 0.033 |
| **Female caregivers' education** | | | |
| Never attend | 1 | 1 | |
| Primary and above | 1.12 [0.83–1.50] | 0.98 [0. 72–1.32] | 0.88 |
| **Male caregivers' education** | | | |
| Never attend | 1 | 1 | |
| Primary and above | 1.46 [1.15–1.85] | 1.42 [1.11–1.81] | 0.005 |
| **Adolescent girls' education** | | | |
| Never attend | 1 | 1 | |
| Primary and above | 2.06 [1.14–3.74] | 1.61 [0.89–2.93] | 0.118 |
| **Male caregiver income** | | | |
| Yes | 1.38 [1.06–1.81] | 1.31 [1.00–1.72] | 0.049 |
| No | 1 | 1 | |
| Adolescent girls age | 0.97 [0.77–1.23] | 1.02 [0.81–1.29] | 0.86 |
| **Household wealth index** | | | |
| Lower | 1 | 1 | |
| Middle | 1.21 [0.90–1.63] | 1.28 [0.95–1.73] | 0.098 |
| Higher | 1.33 [0.99–1.79] | 1.30 [0.96–1.76] | 0.085 |
| **Household food security level** | | | |
| Food secured | 1.54 [1.19–2.0] | 1.33 [1.03–1.74] | 0.031 |
| Food insecure | 1 | 1 | |

**Note:**
The table shows mixed effect logistic regression the DiD interaction term adjusted for other variables.

**Table 5 Heterogeneity in effect of the interventions on family eating practice after adjusting for other variables by adolescent girls' age, and female caregivers' education status, West-Hararghe.**

| | Adolescent girls age | | Female caregivers' education status | |
|---|---|---|---|---|
| | Adolescent age 13 | Adolescent age 14 | Female caregivers have no formal education | Female caregivers with primary and above educational status |
| DID interaction (Time * program) | 2.96 (1.09, 7.99) | 1.41 (0.67, 2.99) | 1.82 (0.88, 3.76) | 4.03 (1.01, 16.13) |
| p-value | 0.032 | 0.369 | 0.105 | 0.049 |
| Observation | 915 | 810 | 1,402 | 323 |

**Note:**
Shows heterogeneity in effect of the interventions on family eating practice after adjusting for other variables by adolescent girls' age, and female caregivers' education status.

## Heterogeneity analyses

We also did a heterogeneity analysis by adolescent girls' ages (13 and 14 separately) and female caregivers' education (never attended and primary and above). After controlling for potential confounders and clustering effect, the DID interaction term showed a significant difference in eating together among the younger adolescents (AOR 2.96 (95% CI [1.09–7.99]), $p$-value of 0.032) and among female caregivers with better educational status (AOR 4.03 (95% CI [1.01–16.13]), $p$-value of 0.049) (Table 5).

## DISCUSSION

The findings of this study showed that the intervention (SAA) improved family eating practices (eating together as a family) that favor adolescent girls after controlling for potential confounding factors such as female caregiver's educational status, adolescent girls' age and education, male caregivers income and household wealth and food security status. The heterogeneity analysis showed that the intervention was more effective among female caregivers with younger adolescent girls (13 years old compared to 14 years) and female caregivers with at least primary level education.

In many societies, food has values attached to it, often related to household power dynamics. Thus, allowing young adolescent girls to eat together with family can promote gender equality and rights within a household, giving girls a fair chance to share what is available (*Alonso, 2014*; *Glover et al., 2019*). In addition, giving young adolescent girls a chance to eat together creates a platform for them to develop healthy eating habits and a sense of belongingness and respect, which are essential for their general well-being (*Heise et al., 2019*; *Baird et al., 2019*; *Kennedy et al., 2020*).

Intrahousehold food allocation social norms directly or indirectly shape family eating practices (*Marcus, 2017*). Families conform with prevailing social norms because they face sanctions and undue pressure from influential community members who are often the norm holders (*Lassi et al., 2017*). Engaging influential community members to deliberate on intrahousehold food allocation norms allows them to understand the negative consequences of the norms, which is a crucial step toward promoting equitable household food allocation that benefits adolescent girls (*Tang et al., 2021*). Furthermore, shifting social norms within a community becomes more feasible when better and acceptable alternatives are offered by the community members (*Cislaghi & Heise, 2018*; *Levy et al., 2020*).

Efforts to shift social norms require understanding the community's health belief systems (*Swigart et al., 2017*; *Lokossou et al., 2021*). The Social Analysis and Action Action (SAA) approach considers perceived barriers and stimulating cues to action in the community. SAA engages norm-holders and influential community members to encourage healthy behavior and influence the belief system (*CARE USA, 2017*). As a result, the community is empowered to overcome the threats of sanctions and adopt positive behaviors. Our finding aligns with the studies that show the effectiveness of a similar intervention in enhancing equitable attitudes, beliefs, and behaviors (*Sedlander et al., 2020*; *Stewart et al., 2021*; *Zimmerman et al., 2021*; *Pichon et al., 2022*). A similar intervention in Kenya to improve social support for mothers' complementary feeding practices also

showed positive results (*Whyte & Kariuki, 1991*; *Mukuria et al., 2016*). Another study in Benin helped to successfully address social norms related to family planning (*Kim et al., 2022*).

The family eating practices could significantly impact health outcomes for adolescent girls and promote family cohesion. The involvement of influential community members is in line with the traditional value given to these groups of people in decision-making in patriarchal societies. Using mixed-effect logistic regression in this study was appropriate as it allowed for the analysis of the clustering effect. Furthermore, the difference in difference analysis employed in this study was critical to control confounding factors. However, the study was conducted in the rural part of the country; thus, the findings may not be generalizable to urban settings and other socio-cultural contexts. Moreover, the study design may not fully control for all potential confounding factors; thus, the influence of residual confounding cannot be ruled out. Finally, the social desirability bias is possible due to the reliance on self-reported family eating practices.

In conclusion, the study indicated that the involvement of female caregivers, along with other influential adult community members, significantly improves the family practice of eating together in households, which is greatly beneficial in promoting equitable allocation of family to adolescent girls. The intervention has great potential to minimize household food allocation inequalities and thus improve the nutritional status of young adolescents if implemented successfully at scale. Further studies are necessary to evaluate the effectiveness of the intervention in different socio-cultural contexts to adopt a policy and guidelines for scale-up.

## ACKNOWLEDGEMENTS

The authors sincerely thank the West Hararge Zone administration and sector offices for their continued support throughout the survey. Also, thanks are extended to all the community members who willingly participated in this study.

### Funding
The authors received no funding for this work.

### Competing Interests
The authors declare that they have no competing interests.

### Author Contributions
- Hanna Gulema conceived and designed the experiments, performed the experiments, analyzed the data, prepared figures and/or tables, authored or reviewed drafts of the article, and approved the final draft.
- Meaza Demissie conceived and designed the experiments, authored or reviewed drafts of the article, and approved the final draft.

- Alemayehu Worku conceived and designed the experiments, performed the experiments, analyzed the data, authored or reviewed drafts of the article, and approved the final draft.
- Tesfaye Assebe Yadeta conceived and designed the experiments, authored or reviewed drafts of the article, and approved the final draft.
- Yemane Berhane conceived and designed the experiments, performed the experiments, analyzed the data, authored or reviewed drafts of the article, and approved the final draft.

## Ethics

The following information was supplied relating to ethical approvals (*i.e.*, approving body and any reference numbers):

Addis Continental Institute of Public Health (Ref No. ACIPH/IRB/005/2016)

## Data Availability

The final dataset used for the analysis is available in the Supplemental File.

## Supplemental Information

Supplemental information for this article can be found online at http://dx.doi.org/10.7717/peerj.16099#supplemental-information.

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
