# Peer review of "Influence of engaging female caregivers in households with adolescent girls on adopting equitable family eating practices: a quasi-experimental study"

_PeerJ, doi:10.7717/peerj.16099_

## Round 0.1 · original submission · Major Revisions

I have received comments from two reviewers and I read this manuscript myself. I think this Manuscript contains some very interesting data, but it should be revised before consideration for publication. Please attempt to revise it in accordance with the all of reviewers' comments.

Reviewer 1 ·

Basic reporting

BASIC REPORTING

 Clear, unambiguous, professional English language used throughout.:
 need some Language editorial issue to improve grammatical issues.
 Intro & background to show context Literature well referenced & relevant.


Introduction:

 The Introduction section needs explanation about the intervention (SAA) component using the current available evidences?
 The introduction needs more explanation on social norm and food allocation that impact to the adolescent girls and the gaps that the research addressed to decision makers. It needs to improve justification of the outcome and link to adolescent girls with food allocation and impact to the adolescent girl.


 line 243 citation needs edition it is not clear? The sentence “Besides, giving young adolescent girls a chance to eat together creates a platform for them to develop healthy eating habits and a sense of belongingness and respect (5) (22,23). Follow the appropriate citation standard according to Peerj.
 line 256 citation needs edition it is not clear? Efforts to shift social norms require understanding the community’s health belief system (29) (30). Follow the appropriate citation standard according to Peerj.

 Structure conforms to PeerJ standards, discipline norm, or improved for clarity: no comment
 Figures are relevant, high quality, well labelled & described

 Line number 195- 210: table 2 overlaps with the paragraph, need rearrange ?

 Raw data supplied (see PeerJ policy). No comment

Experimental design

Original primary research within Scope of the journal.: No comment
 Research question well defined, relevant & meaningful. It is stated how the research fills an identified knowledge gap.

 In method section line 106 -107 indicate as “The group met monthly to discuss, understand, and take action to reduce harmful social norms related to adolescent girls’ health and wellbeing" if the study intervention focused on adolescence girls , need some clarification about the intervention in the introduction section and has some relation for the impact of females' norm to adolescent girls? In the method section the outcome and the study population need additional clarification sentences. Is the study population female care giver or adolescence girls(age 13 and 14)? is the intervention impact for female care giver or adolescent girls age 13 and 14 , make it clear?


 Rigorous investigation performed to a high technical & ethical standard: No comment
 In sampling procedure line number 121 – 122 : “This study analyzed data from households with younger adolescent 122 girls (13-14 years).” Is the analysis adult female care giver or adolescent girls? clearly indicate how female care giver and adolescence girls data linked for this specific study?”

 Methods described with sufficient detail & information to replicate: No comment

Validity of the findings

Impact and novelty not assessed. Meaningful replication encouraged where rationale & benefit to literature is clearly stated.
 The Introduction section needs explanation about the intervention (SAA) component using the current available evidences?
 The introduction needs more explanation on social norm and food allocation that impact to the adolescent girls and the gaps that the research addressed to decision makers. It needs to improve justification of the outcome and link to adolescent girls with food allocation and impact to the adolescent girl.

 All underlying data have been provided; they are robust, statistically sound, & controlled: No comment


 Conclusions are well stated, linked to original research question & limited to supporting results: NO comment

Additional comments

General Comments:

 Language need some editorial issue to improve grammatical issues.

 The results of the study are highly relevant as a public health issue, particularly in the context of cultural norms that have a significant impact on the society specially in developing countries like Ethiopia.


Specific Comments.


Title:

 The tile of the manuscript needs to reflect the study population rather than adult population. which part of the adult population? Is Male, female? it needs to reflect throughout the manuscript with uniform manner.
Abstract:

 In the abstract section on line 38-39 “We used data from a large quasi-experimental study to compare family eating practices between those who participated in the intervention (SAA) and those who did not” The abbreviation SSA is not clear? it needs some explanation about it , since abstract is self-contained and clear for readers. It needs clear about the intervention components in short rather than explaining using backer OR the intervention component.
 The outcome of the manuscript indicated clearly in line 41 and 42 as “The study’s outcome was the practice of family eating together from the same dish” It is not indicated in the main method section of the manuscript. It needs to clearly indicate and explained in the method/measurement section.

Introduction:

 The Introduction section needs explanation about the intervention (SAA) component using the current available evidence?
 The introduction needs more explanation on social norm and food allocation that impact to the adolescent girls and the gaps that the research addressed to decision makers. It needs to improve justification of the outcome and link to adolescent girls with food allocation and impact to the adolescent girl.


Methods:


 In method section line 106 -107 indicate as “The group met monthly to discuss, understand, and take action to reduce harmful social norms related to adolescent girls’ health and wellbeing" if the study intervention focused on adolescence girls, need some clarification about the intervention in the introduction section and has some relation for the impact of females' norm to adolescent girls? In the method section the outcome and the study population need additional clarification sentences. Is the study population female care giver or adolescence girls (age 13 and 14)? is the intervention impact for female care giver or adolescent girls aged 13 and 14 , make it clear?

 In sampling procedure line number 121 – 122 : “This study analyzed data from households with younger adolescent 122 girls (13-14 years).” Is the analysis adult female care giver or adolescent girls? clearly indicate how female care giver and adolescence girls' data linked for this specific study?”

Result:


 Line number 195- 210: table to overlaps with the paragraph, need rearrange?


Discussions:

 line 243 citation needs edition it is not clear. The sentence “Besides, giving young adolescent girls a chance to eat together creates a platform for them to develop healthy eating habits and a sense of belongingness and respect (5) (22,23). Follow the appropriate citation standard according to Peerj.
 line 256 citation needs edition it is not clear. Efforts to shift social norms require understanding the community’s health belief system (29) (30). Follow the appropriate citation standard according to Peerj.

Conclusion:

 conclusion section is well written part of the manuscript.

References:


 Line 378 “Rachel Marcus. Advancing Learning and Innovation on Gender Norms. 2018 Aug.” What kind of reference is it? is the citation included all components that should be indicated in the reference list?

Annotated reviews are not available for download in order to protect the identity of reviewers who chose to remain anonymous.

·

Basic reporting

Overall, it’s a very interesting study but it would benefit greatly from a more thorough and clear presentation. I believe there is a lot of potential in this study however, it is not yet at a stage to be revised and published - therefore, I recommend a reject and resubmit.

1. The authors do a fine job of starting to organize their study design and data collection. A lot more information on the set-up is needed though. For example-
a) How many kebeles in West Hararghe received the intervention?
b) I am curious about implementation fidelity. Given that this was a pretty long intervention, what was the take-up of the program initially and then attendance for those who were treated.
c) Clarity on the evaluation is needed- it sounds like both arms received some form of programming but only the “eating practices” are being evaluated? I would like to see more details on this intervention to really connect with the outcomes. Perhaps a schematic description would be useful.

Experimental design

2. This analysis really needs some sort of a treatment-control comparison table. Either a baseline balance table (if random assignment) or parallel trends assumption (if using DID) or common support (if using matching).

3. Motivation for the choice of model is unclear. If treatment is not randomly assigned, why not use a matching design?
a. What was the level of clustering?
b. What are the covariates that were unbalanced?
c. Proper justification of these terms and how they affect causality is needed.
d. Proper subscripts for the algebraic terms are needed

Validity of the findings

4. Result presentation- The tables need a lot of clean-up and proper discussion.
a. Table 3 is missing a careful discussion, given it’s the main result
b. I would refrain from using confidence intervals and just report standard errors or p-values. It makes the tables difficult to understand.
c. It would be very helpful to format the results as in other published articles.
d. Please give extensive notes to tables and refer to other peer-reviewed articles for stylizing.

Additional comments

5. Again, this seems like a study with a lot of potential but not yet journal quality. Some suggestions to enrich beyond the ones above are-
a. Expand on the list of outcomes like for example, nutritional impacts on adolescents, number of meals consumed etc.
b. Have a section on mechanisms
c. Have some interesting heterogeneity analysis

---

## Round 0.2 · accepted · Accept

Please work with the Editorial office to correct the parts about references pointed out by Reviewer 1.

Reviewer 1 ·

Basic reporting

All comments have been incorporated and corrected in the basic report section of the manuscript. No comment on this section.

Experimental design

no comment, The author has incorporated all my comments into the Experimental design section of the manuscript.

Validity of the findings

no comment

Additional comments

In response to all my comments, the author addressed all my comments. Except some minor reference comments.

Annotated reviews are not available for download in order to protect the identity of reviewers who chose to remain anonymous.

·

Basic reporting

Checked

Experimental design

Checked

Validity of the findings

Checked

Additional comments

Checked